

# Widely targeted metabolite profiling of mango stem apex during floral induction by compond of mepiquat chloride, prohexadione-calcium and uniconazole

Fei Liang[1,2,*], Wentian Xu[1,*], Hongxia Wu[1], Bin Zheng[1], Qingzhi Liang[1], Yingzhi Li[2] and Songbiao Wang[1]

[1] Key Laboratory of Tropical Fruit Biology of Ministry of Agriculture, South Subtropical Crops Research Institute, Chinese Academy of Tropical Agricultural Sciences, Zhanjiang, China
[2] Binhai Agricultural College of Guangdong Ocean University, Zhanjiang, China
[*] These authors contributed equally to this work.

Corresponding authors
Yingzhi Li, liyz@gdou.edu.cn, gdliyingzhi@yeah.net
Songbiao Wang, wsbcjy@163.com

## ABSTRACT

**Background**. Insufficient low temperatures in winter and soil residues caused by paclobutrazol (PBZ) application pose a considerable challenge for mango floral induction (FI). Gibberellin inhibitors SPD (compound of mepiquat chloride, prohexadione-calcium and uniconazole) had a significant influence on enhancing the flowering rate and yield of mango for two consecutive years (2020–2021). Researchers have indicated that FI is regulated at the metabolic level; however, little is known about the metabolic changes during FI in response to SPD treatment.

**Methods**. Here, ultra-performance liquid chromatography-electrospray ionization-tandem mass spectrometry (UPLC-ESI-MS/MS)-based widely targeted metabolomic analysis was carried out to assess the metabolic differences in the mango stem apex during different stage of mango FI (30, 80, 100 days after SPD/water treatment).

**Results**. A total of 582 compounds were annotated and 372 metabolites showed two-fold differences in abundance (variable importance in projection, VIP ≥ 1 and fold change, FC≥ 2 or≤ 0.5) between buds at 30, 80, 100 days after SPD/water treatment or between buds under different treatment. Lipids, phenolic acids, amino acids, carbohydrates, and vitamins were among metabolites showing significant differences over time after SPD treatment. Here, 18 out of 20 lipids, including the lysophosphatidylethanolamine (12, LPE), lysophosphatidylcholine (7, LPC), and free fatty acids (1, FA), were significantly upregulated from 80 to 100 days after SPD treatment comared to water treatment. Meanwhile, the dormancy release of mango buds from 80 to 100 days after SPD treatment was accompanied by the accumulation of proline, ascorbic acid, carbohydrates, and tannins. In addition, metabolites, such as L-homocysteine, L-histidine, and L-homomethionine, showed more than a ten-fold difference in relative abundance from 30 to 100 days after SPD treatment, however, there were no significant changes after water treatment. The present study reveals novel metabolites involved in mango FI in response to SPD, which would provide a theoretical basis for utilizing SPD to induce mango flowering.

## INTRODUCTION

Mango (*Mangifera indica* L.) is one of the most important fruit crops in the tropical and subtropical regions (*He et al., 2021*). China is the second largest mango production country (*Li et al., 2019*). Floral induction (FI) is an essential development event in perennial woody fruit trees, which determines the onset of fruit and plays a crucial role in fruit yield (*Wilkie, Sedgley & Olesen, 2008*). Mango FI starts from dormant bud or growing bud and ends with the beginning of flower bud morphological differentiation (commonly known as brush head in production). Although the mechanisms that control flowering remain primarily unknown in mango, it can be induced by low temperatures (*Davenport, 2007*). However, insufficient cold winter temperatures due to global warming have affected mango FI and resulted in reduced fruit production in commercial mango orchards in China, mainly in Hainan, Guangdong and Guangxi.

In recent decades, soil drenching with paclobutrazol (PBZ) has been used as a major practice to induce mango flowering in many commercial mango orchards, including Guangdong, Guangxi and Hainan in China. PBZ inhibits gibberellin biosynthesis and vegetative growth, resulting in mango FI (*Guevara, Jiménez & Bangerth, 2012*). However, excessive doses of PBZ over a long period can result in soil residues (*Milfont et al., 2008*; *Silva et al., 2017*), new bud and panicle compaction, and increased disease occurrence (*Singh & Bhattacherjee, 2005*; *Coelho, 2014*). Specifically, mango inflorescence malformation damages the mango industry and threatens mango productivity (*Freeman et al., 2014*; *Ansari et al., 2015*). It also strongly affects the normal growth of plants and human health (*Jiang et al., 2019*; *Guo et al., 2020*). In addition, PBZ has a low mobility in the soil, so plants cannot fully absorb it, and therefore it will remain in the soil for a long time. It will lead to pollution of surface water and groundwater resources (*Sharma & Awasthi, 2005*; *Silva et al., 2017*). In addition to PBZ, there are several safer gibberellin inhibitors such as uniconazole, prohexadione calcium and mepiquat chloride which play roles in regulating plant growth. Uniconazole can inhibit the activity of ent-kaurene oxidase (KO) in gibberellin biosynthesis and promote flowering of juvenile macadamia trees (*Nagao, Ho-a & Yoshimoto, 1999*; *Rademacher, 2000*). Prohexadione calcium hinders gibberellin (GA) biosynthesis to retard the runner formation of strawberry (*Hytönen et al., 2009*; *Kim et al., 2019*). Mepiquat chloride promotes cotton lateral root formation by inhibiting GA biosynthesis and signal transduction (*Wu et al., 2019*). More importantly, they have the characteristics of low toxicity and low residue in soil (*Basak & Rademacher, 2000*; *Bazzi et al., 2003*; *Li et al., 2012*; *Dong, Zhu & Chen, 2021*; *Huang et al., 2021*). The effect of uniconazole is 4–10 times higher than that of paclobutrazol, but its residue in soil is only 1/10 of that of paclobutrazol (*Gilbertz, 1992*; *Barrett & Nell, 1992*; *Sellmer et al., 1999*). Prohexadione calcium can be rapidly degraded into water and carbon dioxide by microorganisms in the soil, and has no residual toxicity to rotation plants and no pollution to the environment (*Evans et al., 1998*; *Ilias & Rajapakse, 2005*). Whereas, the effects of these agents on mango flowering regulation have not been evaluated.

Studies have revealed that specific metabolites are involved in FI in perennial fruit trees. However, these studies mainly focused on the carbohydrates and endogenous hormones in

leaves and flower buds and their changes during FI (*Upreti et al., 2013*; *Upreti et al., 2014*; *Xing et al., 2015*). Research has also evaluated the effects of metabolites such as sucrose and fulvic acid with PBZ on FI (*Du et al., 2017*; *dos Santos Silva et al., 2021*). Meanwhile, early reports in mango showed that other metabolites, such as amino acids and phenols, were involved in mango FI (*Tiwari, Patel & Pandey, 2018*). *Osuna-Enciso et al. (2001)* reported higher amino acids content in mango flower buds than in leaf buds, indicating the increased need for amino acid during flowering (*De Postgraduados et al., 2001*; *Zhang et al., 2022*). Meanwhile, in Arabidopsis, the increase in phosphatidylcholine (PC) levels in the stem meristems accelerated flowering while the decrease in PC levels in the stem meristems delayed flowering, indicating a correlation between PC levels and the flowering time (*Nakamura et al., 2014*; *Nakamura et al., 2019*). However, the metabolic changes during the different stages of mango FI, determined by a metabolomics approach, have not been reported.

Metabolomics is an emerging 'omics' tool following genomics, transcriptomics, and proteomics. Researchers have used widely targeted metabolite analysis based on tandem mass spectrometry (MS/MS) data for large-scale metabolite profiling in various plant species (*Wang et al., 2018a*; *Wang et al., 2021*; *Zou et al., 2020*; *Yi et al., 2021*; *Xiao et al., 2021*). Plant primary metabolites, including sugars, amino acids, lipids, nucleotides, and other substances, act as essential substances and energy sources for plant growth, development and reproduction. Based on the ultra-performance liquid chromatography-tandem mass spectrometry (UPLC-MS/MS) detection technology, a secondary spectral matching method was developed for the qualitative and multiple reaction monitoring (MRM) of substance relative content, using an independently constructed plant primary metabolite database. The metabolites were evaluated using both univariate and multivariate statistical methods.

In this study, we evaluated the effects of SPD, a compound chemical composed of uniconazole, prohexadione calcium and mepiquat chloride, on the flowering and yield of mango, and we recorded the morphological characteristics of buds at different days (30, 60, 80, 100) after SPD/water treatment. In addition, a UPLC-MS/MS analysis was performed to obtain a comprehensive and dynamic metabolic profile, identify specific metabolites, and explore the critical metabolic pathways in response to SPD treatment during mango FI. The findings will provide a potential method for mango flowering management in subtropical regions and provide a theoretical basis for applying SPD to regulate mango flowering.

# MATERIALS & METHODS

## Plant material

The experimental orchard is a commercial orchard, and the orchard adopts conventional management measures. Except for a brief temperature drop in mid-December (below 15 °C for about one week), the average monthly temperature for the other months is higher than 15 °C. Before flowering, branches were not pruned and flies were bred in the orchard to pollinate. In our pre-experiment, we sprayed about 12 L of water on the leaves of nine

trees. Eighteen randomly selected fifteen-year-old 'Tainong' mango trees (*Mangifera indica* L.) grown at the commercial orchard at the South Subtropical Crops Research Institute of the Chinese Academy of Tropical Agricultural Science in Zhanjiang, China (110°16′E, 21°10′N) were used in this study from 2019 to 2021. In late September of years 2019 and 2020, when the leaves of the second flush turned green, nine trees were selected and their leaves were sprayed with 12 L of SPD (mepiquat chloride: 2 g L-1, prohexadione-calcium: 100 mg L-1, uniconazole: 300 mg L-1 (researchers developed the formula for exogenous gibberellin inhibitors.) or 12 L of water (as a control) on the leaves of each tree, respectively. All other measures are routine management measures. The experiments were conducted in a randomized block design with three repetitions per treatment and three trees per repetition.

According to the classification of mango phenological growth stages described by *Rajan et al. (2011)* and *Ramírez et al. (2014)*, researchers still do not know the detail information about the phonological changes from dormant bud to flower bud initiation stage. Thus, morphological changes in the bud are regularly observed and photographed until the bud is released. We also refer to *Yang et al. (2021)* for making paraffin sections and observing them (*Oliveira et al., 2020*). Stage 1 is the dormant stage when the buds are dormant, about 30–60 days after SPD/water treatment. Stage 2 is the green tip stage, about 60–80 days after SPD/water treatment. During this stage, the swelling buds expose the inner leaf primordia under the SPD treatment but still dormant buds under the water treatment. Stage 3 is bud initiation stage, about 80–100 days after SPD/water treatment. At this stage, the apex of the mango stem is covered by elongated bracts when treated with SPD, but the buds remain dormant until 100 days after water treatment. After that, buds entered the stage of morphological differentiation according to *Oliveira et al. (2020)* under the SPD treatment but were still in dormancy under the control. Terminal buds were collected at three different stages shown in Fig. 1 (30, 80 and 100 days after SPD/water treatment) from October 2020 to January 2021. Experiments were carried out on three stages of buds with three biological replicas. Each biological replica consists of a pool of 12 buds from three trees. The bud samples collected were immediately placed in liquid nitrogen and stored in a freezer at −80 °C.

We tagged a total of 20 branches evenly distributed on the east, west, north and south sides of each tree, and we recorded the branches with flowers as flowering branches for each tree. About 140 days after SPD/water treatment, the flower formation rate was evaluated using the ratio of the number of flowering branches in the entire tree to the total number of branches in each tree. When the fruit reached commercial harvest maturity, we calculated the mango yield per tree. There are nine trees for SPD and water treatment respectively.

## Metabolite detection

Metabolite extraction: Bud samples were pre-treated and extracted as previously described (*Wang et al., 2018b*). Briefly, the frozen bud samples mentioned above were lyophilized and smashed with zirconia beads in a mixer mill (MM 400; VERDER RETSCH, Shanghai, China) for 1.5 min. Then, 0.1 g of sample powder was extracted in 1.2 mL 70% methanol solution overnight at 4 °C. The supernatant obtained after centrifugation at 12,000

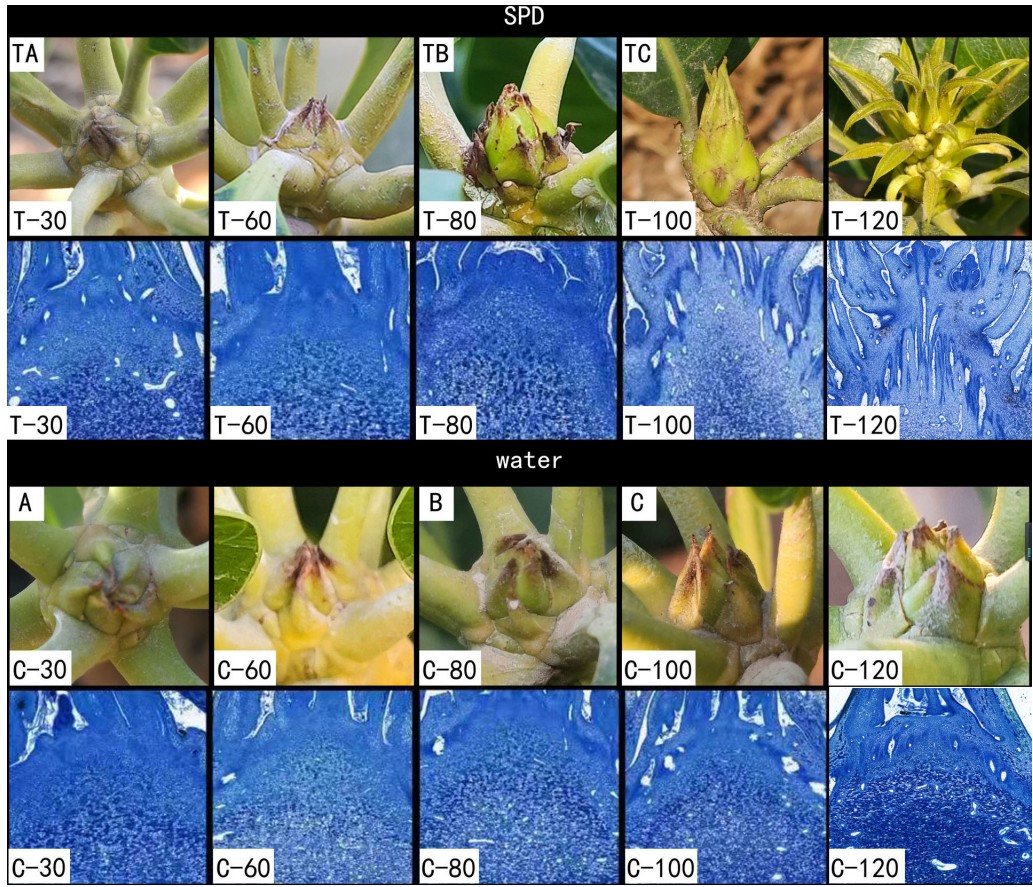

**Figure 1** **Morphological changes of mango buds at different stages of floral induction and initiation.** TA, TB, TC, A, B and C indicates sample names. T-30/C-30 indicates mango stem apex at 30 days after SPD/water treatment. T-60/C-60 indicates mango stem apex at 60 days after SPD/water treatment. T-80/C-80 indicates mango stem apex at 80 days after SPD/water treatment. T-100/C-100 indicates mango stem apex at 100 days after SPD/water treatment. T-120/C-120 indicates mango stem apex at 120 days after SPD/water treatment.

rpm for 10 min was passed through a CNWBOND Carbon-GCB SPE Cartridge (250 mg, three mL; ANPEL, Shanghai, China) and filtered (0.22 µm pore size; SCAA-104). L-2-chlorophenylalanine was used as the standard solution in the study.

UPLC conditions: The bud extract was analyzed using an ultra-performance liquid chromatography-electros pray ionization-tandem mass spectrometry (UPLC-ESI-MS/MS; Shanghai, China) system (UPLC, SHIMADZU NexeraX2; MS, 4500 Q TRAP Applied Biosystems, Waltham, MA, USA). First, the aliquot (4 µL) was injected into the Agilent SB-C18 column (1.8 µm, 2.1 mm ×100 mm) in an UPLC system. During sample analysis, the composition and the ratio of mobile phases were changed slightly (*Zou et al., 2020*). The separation was achieved using the mobile phases solvent A (pure water with 0.1% formic acid) and solvent B (acetonitrile with 0.1% formic acid) and a gradient program as follows: the starting conditions of 95% A, 5% B. Within 9 min, a linear gradient to 5% A, 95% B was programmed, and a composition of 5% A, 95% B was kept for 1 min.

Subsequently, a composition of 95% A, 5.0% B was adjusted within 1.10 min and kept for 2.9 min. The flow velocity was set at 0.35 ml per minute. The column oven was set to 40 °C.

ESI-Q TRAP-MS/MS: The effluent was alternatively connected to an AB4500 Q TRAP UPLC/MS/MS system, equipped with an ESITurbo Ion-Spray interface, operating in both positive and negative ionmodes and controlled by Analyst software (v1.6.3; AB Sciex, Toronto, Canada). The operating parameters were similar to those in the previous study (*Zhang et al., 2020*). An ion source temperature of 550 °C and ion spray voltage (IS) of 5500 V (positive ion mode)/−4500 V (negative ion mode) were used; ion source gas I (GSI), gas II (GSII), and curtain gas (CUR) were set at 50, 60, and 25.0 psi, respectively. Finally, a specific set of MRM transitions were monitored for each period according to the metabolites eluted within this period.

## Qualitative and quantitative analyses of metabolites

Qualitative analysis: The MS data were analyzed based on the self-built database of purified metabolite standards and the public metabolite database in collaboration with the MWDB (MetWare Biotechnology Co., Ltd., Wuhan, China). The accurate precursor ion (Q1) and production (Q3) values, retention times, and fragmentation patterns were compared with those of the standards to analyze the primary and secondary MS information, followed by the removal of few repeated signals (*Zheng et al., 2021*).

Quantitative analysis: The quantitative analysis was carried out using the MRM mode of the triple quadrupole (QQQ) MS. The ions corresponding to other molecular weight substances were initially excluded to avoid interference, and the parent ions of the target substances in the MRM mode were searched, leading to higher precision and repeatability of results.

The peak area of all substance mass peaks was integrated after metabolite mass spectrometry data from several samples were obtained, and the peaks of the same metabolite in various samples were integrated and corrected. The relative content of metabolites is the integral value of the peak area. Quality control samples (QC) samples are prepared by mixing extracts from sample samples and are used to analyze sample repeatability within the same treatment environment. A quality control sample is inserted into every ten test analysis samples during the analysis to ensure that the process is repeatable.

## Statistical analysis

The peak areas for each metabolite were normalized by unit variance (UV) scaling. The normalized metabolite data of all samples were analyzed using principal component analysis (PCA) by the statistics function of prcomp within R (v3.63) (http://www.r-project.org), hierarchical cluster analysis (HCA) by the pheatmap in R package (v1.0.12). The relative abundance of all differential metabolites was normalized by z-score, followed by K-Means clustering analysis by R, and orthogonal projections to latent structures-discriminant analysis (OPLS-DA) by MetaboAnalystR (v1.0.1) package in R (*Thévenot et al., 2015*) were used to evaluate the metabolic differences between and within bud samples. The variable importance in the projection (VIP) in the OPLS-DA model was used to identify the differential metabolites (VIP ≥ 1 and absolute log2FC|fold change|≥1). Annotated

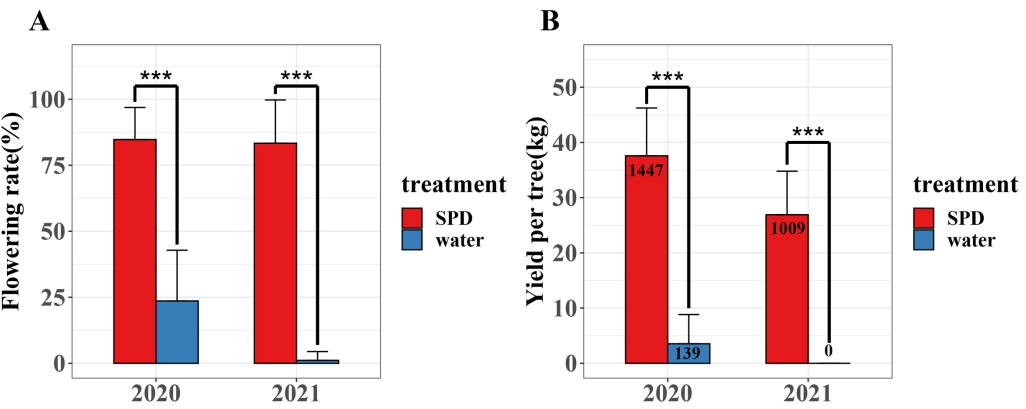

**Figure 2   Flowering rate and yield per tree were after SPD treatment or water (control) in 2020 and 2021.** (A) Flowering rate; (B) Mango yield per tree. Note: an asterisk (*) indicates statistical significance with $p < 0.05$, two asterisks (**) indicate statistical significance with $p < 0.01$, three asterisks (***) indicate statistical significance with $p < 0.001$. The numbers in figure represent the number of mango.

metabolites were mapped to the KEGG database (http://www.kegg.jp/kegg/pathway.html) to determine the pathway associations. Metabolite set KEGG enrichment analysis was performed on the pathways enriched with significantly regulated metabolites; the pathways with Bonferroni corrected $P$-values $\leq 0.05$ were considered significantly enriched.

## RESULTS

### SPD treatment enhanced flowering rate and yield

We investigated flowering rate and yield per tree of mango trees treated with SPD and water for two consecutive years (2020–2021) (Fig. 2 and Table S1). In 2020 and 2021, the flowering rate and mango yield per tree of SPD treatment were significantly higher than those of the control, and the flowering rate for two consecutive years exceeded 80%. In 2021, the number and yield of mangos per tree treated by SPD decreased slightly compared with that in 2020, which may be due to some other factors, such as unfavourable temperature and rainfall. The results confirmed the efficiency of SPD on mango flower induction.

### Morphological characteristics of buds in response to SPD treatment

Both environmentally induced dormant bud and the growing bud can initiate floral bud in mango (*Batten & Mcconchie, 1995*). Growing points of mango buds were covered by green or brownish scales, which is different from the buds of some tropical evergreen perennials, such as litchi, which are naked (*Zhang et al., 2016*). With the maturation of the second flush, the dormant buds were thin and covered with green scales that tightly embraced each other, and the stem apices were concealed during 30 and 60 days after treatment with SPD/water (Fig. 1 T-30/C-30, T-60/C-60). Then, the buds started swelling and exposing the inner leaf primordia at 80 days after SPD treatment, but no obvious changes were observed in buds treated with water (Fig. 1 T-80/C-80). Finally, the scales separated, and the base of mango bud is enlarged and the top is pointed (a situation that indicates potential inflorescence formation) at 100 days after SPD treatment, but scales of the bud only slightly

loosened under the water control (Fig. 1 T-100/C-100). After that, the first floral primordia were visible, and the panicles started to develop at 120 days after SPD treatment, whereas, scales of the bud treated with water were still mildly separated (Fig. 1 T-120/C-120). In order to further confirm the flower bud differentiation stage under SPD treatment, the terminal buds of 30 days, 60 days, 80 days, 100 days, and 120 days after SPD treatment were analyzed by histological cytology. The stem apex meristem was dome-shaped at 30–60 days after water/SPD treatment, then expanded laterally and lost its dome shape at 80 days after water/SPD treatment. 100 days after SPD treatment, inflorescence primordia appeared in the stem apex of the major axis and lateral axis, displaying a typical inflorescence structure, a sign of flower bud morphological differentiation stage. The stem apex of water treatment remained in the dormant stage.

During this morphogenic process, a targeted metabolomic analysis was conducted by collecting mango buds at different stages of induction, *e.g.,* at 30 days after treatment with SPD/water (TA/A), 80 days after treatment with SPD/water (TB/B), and 100 days after treatment with SPD/water (TC/C), respectively, to investigate the metabolic dynamics during FI.

## Metabolic characteristics of buds at different days in response to SPD treatment

In total, 582 metabolites were annotated from the mango buds at 30, 80, 100 days after SPD/water treatment (Table S2). A PCA was conducted on the 582 metabolites, with PC1 and PC2 explaining 39.62% and 19.31% of the total variation, respectively. Lipids and phenolic compounds are the main contributions to PC1, such as LysoPC 18:3, LysoPC 18:2, LysoPC 16:2, Methyl gallate, Digallic acid, 1,3,6-Tri-O-galloyl- $\beta$-D-glucose, 4-O-Methylgallic Acid and L-Malic acid-2-O-gallate. Phenolic compounds and organic acids are the main contributions to PC2, such as 2-hydroxycinnamic acid, 3-hydroxycinnamic acid, trans-5-O-(p-coumaroyl)shikimate, phenyl acetate, 2-hydroxy-2-methyl-3-oxobutanoic acid, 2-methylsuccinic acid, 4-hydroxy-2-oxopentanoic acid and monomethyl succinate. The 18 bud samples from three stages of two treatments distributed into distinct clusters, implying differences in the metabolic characteristics among the bud at 30, 80, 100 days after SPD/water treatment. It is consistent with the change of phenotypic characteristics (Fig. 3A). Besides, the r values (correlation coefficient) of the biological repeats were greater than 0.97, while the correlation between the different samples was low, indicating relatively good sample repeatability (Fig. 3B). Further, a log10 transformation of peak area was applied to each metabolite to eliminate the effects of quantity on pattern recognition, and an HCA analysis was performed. This analysis revealed six distinct groups associated with TA, TB, TC, A, B and C, respectively. The content of metabolites in groups 1 and 2 was the highest in samples of 100 days after water treatment; that in groups 3 and 4 was the highest in samples of 30 or 80 days after water treatment; that in groups 5 was the highest in samples of 80 and 100 days after SPD treatment; and that in groups 6 was the highest in samples of 100 days after SPD/water treatment (Fig. 3C). Thus, the PCA and HCA analysis revealed differences in various metabolite spectrums among bud samples of different stages and treatments with good reproducibility and stability.

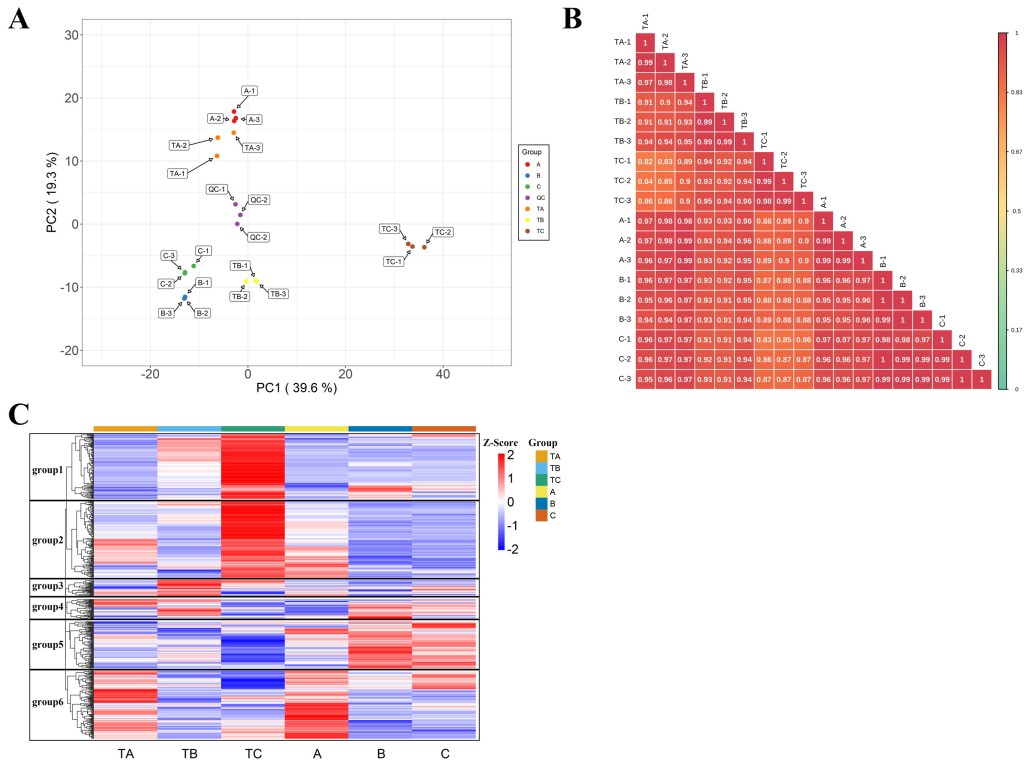

**Figure 3** **Metabolic characteristics of buds at different days.** (A) Principal component analysis of metabolites TA, TB, TC, A, B and C. PC1 represents the first principal component, PC2 represents the second principal component, and the percentage represents the interpretation rate of the principal component of the data set. Each dot in the diagram represents a sample, and samples from the same group are represented in the same color; TA-1, TA-2 and TA-3 are three repetitions of TA. TB-1, TB-2 and TB-3 are three repetitions of TB. TC-1, TC-2 and TC-3 are three repetitions of TC. A-1, A-2 and A-3 are three repetitions of A. B-1, B-2 and B-3 are three repetitions of B. C-1, C-2 and C-3 are three repetitions of C. QC-1, QC-2 and QC-3 are three repetitions of QC. (B) Sample correlation analysis of mango buds during Fl based on relative amounts of metabolite contents. stem apex in three stages (treated with SPD/water), the abscissa and ordinate represent the sample name, and the color represents the correlation value (red represents high, green represents low). (C) Hierarchical cluster analysis of metabolites from samples of TA, TB, TC, A, B and C. The horizontal represents the metabolites, the vertical represents the sample name, and the different colors are the values obtained by standardization of the relative amounts (red represents high, blue represents low).

Subsequently, the metabolites showing an FC ≥ 2 (up-regulated) or ≤ 0.5 (down-regulated) between the A and B, the A and C, the B and C, the TA and TB, the TA and TC, the TB and TC, the A and TA, the B and TB, and the C and TC were selected for further analysis to identify the differential metabolites in the mango buds at different days after SPD/water treatment or in buds under different treatments. The VIP value from the OPLS-DA model (VIP ≥ 1) was used to screen these metabolites, and a total of 372 metabolites were identified from the nine comparisons. Among these, 143 differential metabolites (36 up-regulated and 107 down-regulated) were identified between A and B; 105 differential metabolites (23 up-regulated and 82 down-regulated) were identified between A and C; 20 differential metabolites (13 up-regulated and 7 down-regulated) were identified between

B and C; 131 differential metabolites (73 up-regulated and 58 down-regulated) were identified between TA and TB; 151 (113 up-regulated and 38 down-regulated) between TA and TC; and 224 (155 up-regulated and 69 down-regulated) between TB and TC, 22 differential metabolites (seven up-regulated and 15 down-regulated) were identified between A and TA, 101 differential metabolites (72 up-regulated and 29 down-regulated) were identified between B and TB, 252 differential metabolites (179 up-regulated and 73 down-regulated) were identified between B and C (Fig. 4A). Furthermore, there were 14 differential metabolites constantly altered by water treatment, and 53 differential metabolites continuously altered by SPD treatment. Compared with the control, seven common metabolites were found in buds at 30, 80 and 100 days after SPD/water treatment (Figs. 4B–4D).

## KEGG classification and enrichment analysis of differential metabolites

Furthermore, the differential metabolites (between A and TA: 22, B and TB: 101 and C and TC: 252) were mapped to the KEGG database. The results demonstrated that most of the metabolites were related to the metabolism and biosynthesis of secondary metabolites. Few metabolites were associated with the biosynthesis of amino acids. The remaining pathways included ABC transporters, 2-oxocarboxylic acid metabolism, and aminoacyl-tRNA biosynthesis, only the least proportion of metabolites were represented. Subsequently, KEGG enrichment analysis of the differential metabolites was done to determine the differences in metabolic pathways between the A and TA stages, the B and TB stages, the C and TC stages. The enrichment analysis showed that the metabolites derived from glycine serine and threonine metabolism, cysteine and methionine metabolism, zeatin biosynthesis, biosynthesis of amino acids, glycerolipid metabolism, tryptophan metabolism, phenylalanine tyrosine and tryptophan biosynthesis. Carbon fixation in photosynthetic organisms and arginine and proline metabolism were significantly different between the B and TB stages, the C and TC stages ($p < 0.05$) (Fig. 5 and Table S3).

## K-means clustering analysis of differential metabolites

The average of three repeated relative abundances of all differential metabolites in each sample was standardized by z-score and analyzed by K-means clustering to obtain the accumulation pattern of the differential metabolites across the different stages of mango FI (Fig. 6 and Table S4). The 372 differential metabolites (125 phenolic acids, 78 lipids, 52 amino acids and derivatives, 36 nucleotides and derivatives, 42 organic acids, and 39 others) were divided into six clusters. Among them, the phenolic acids were classified into the cluster 2, 3, 4, 6 and 8 (12.8%, 12.8%, 12.8%, 12.8% and 19.2%), lipids were classified into the cluster 4 and 7 (26.9% and 33.3%), amino acids and derivatives were mainly classified into the cluster 4 and 6 (32.6% and 26.9%), nucleotides and derivatives were classified into the cluster 4 (30.5%), organic acids were classified into the cluster 1 and 3 (16.7% and 19%), and others were classified into the cluster 3 and 6 (33.3% and 25.6%).

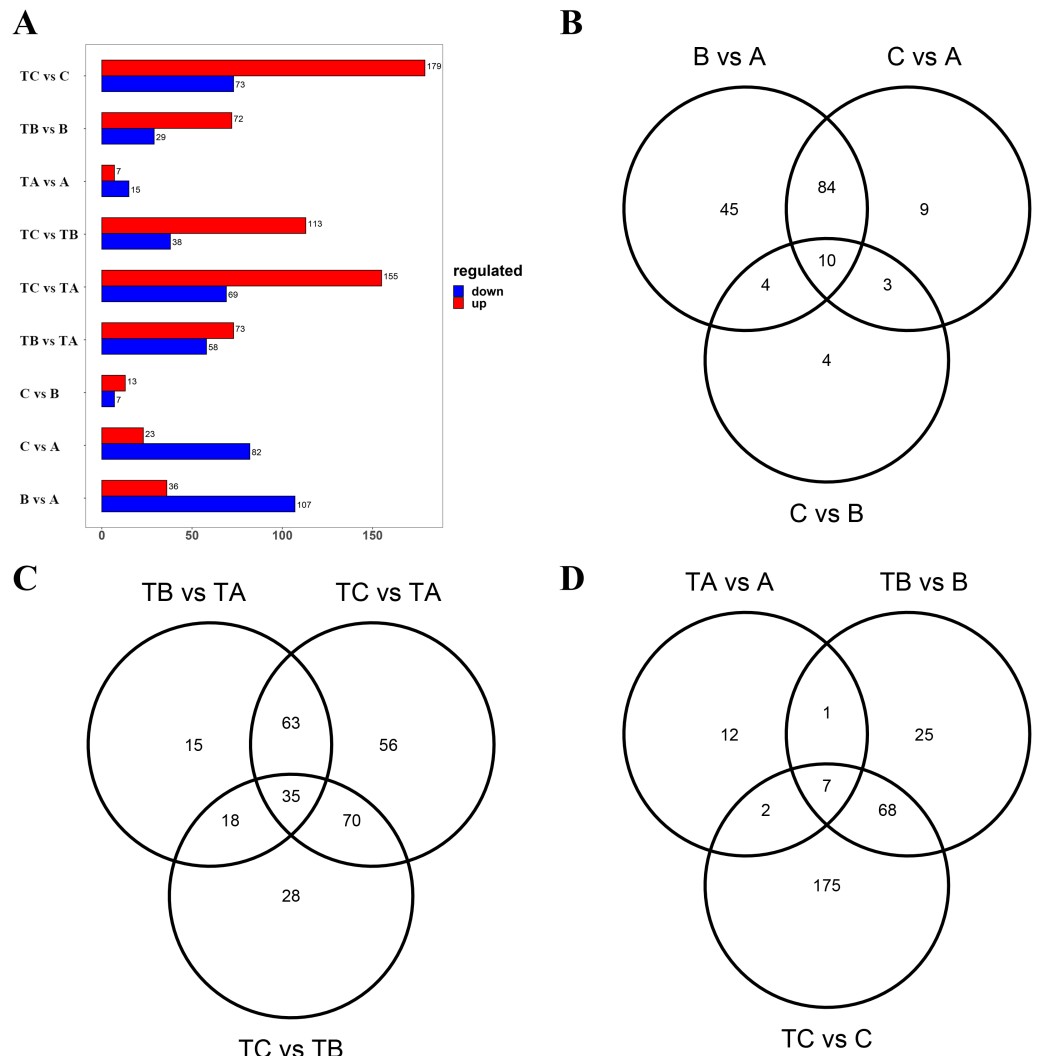

**Figure 4 Differential metabolites between different groups.** (A) Differential metabolites between different sample. The abscissa indicates the quantity of differential metabolites, the ordinate indicates different groups, and red represent up-regulated differential metabolites, and blue represent down-regulated differential metabolites. And up regulated / down regulated is the change of the former relative to the latter. (B) Venn diagram of differential metabolites B *vs* A, C *vs* A, and C *vs* B. (C) Venn diagram of differential metabolites TB *vs* TA, TC *vs* TA, and TC *vs* TB. (D) Venn diagram of differential metabolites TA *vs* A, TB *vs* B, and TC *vs* C.

## Identification of lipids, phenolic acids, and amino acids during FI after SPD treatment

To select metabolites that might be related to FI instead of SPD treatment, we focused on metabolites with significant differences during FI in response to SPD treatment, comparing their accumulation patterns after different days both by SPD and water treatment. The present study identified 53 metabolites (Table S5) during FI that showed significant changes from TA to TB and TB to TC after SPD treatment, including 20 lipids, 18 phenolic acids, seven amino acids and their derivatives, five nucleotides and their derivatives, two organic

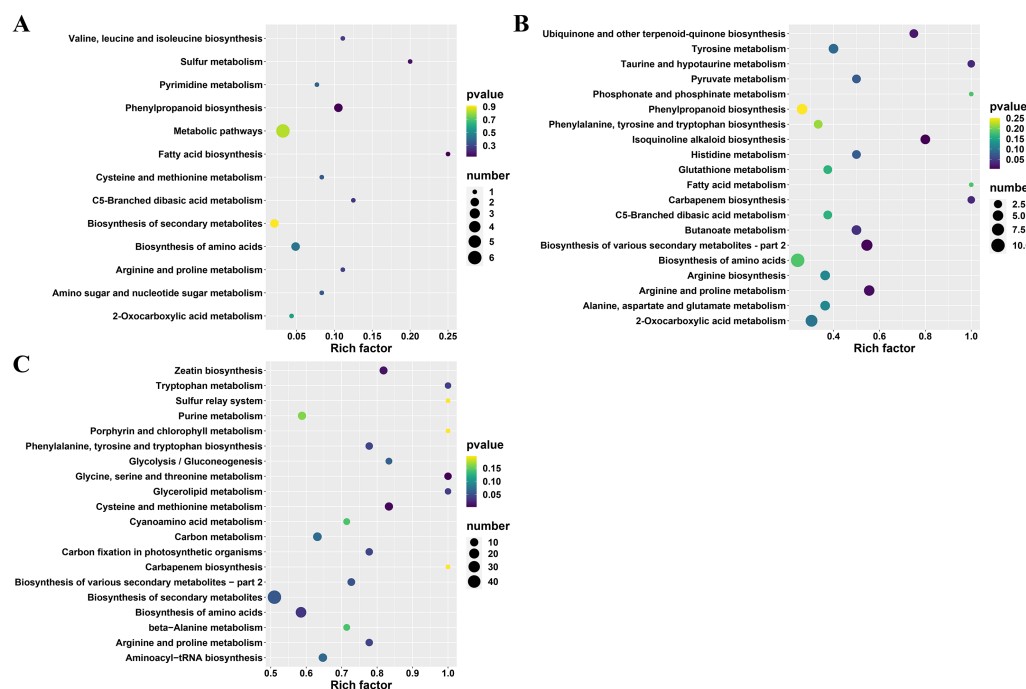

**Figure 5  KEGG enrichment analysis of differential metabolites.** (A) TA *vs* A, (B) TB *vs* B, and (C) TC *vs* C; The abscissa indicates the rich factor corresponding to each channel, the ordinate indicates the channel name, and the color of the point is *P* value. The size of the dot represents the number of enriched differential metabolites.

acids, and 1 other (Fig. 7A). The results suggested that these metabolites, especially lipids, phenolic acids, and amino acids, might play a key role in FI in respond to SPD treatment in mango. 20 differently changed lipids included 12 lysophosphatidylethanolamine, seven lysophosphatidylcholine, and one free fatty acid. A total of 18 out of the 20 lipids showed a significant decrease towards TB and a substantial increase towards TC following SPD treatment, but a significant decrease towards B and no alterations towards C after water treatment. The result indicated the significance of lipids decomposition and synthesis during mango FI by SPD treatment. Meanwhile, 12 out of 18 phenolic acids significantly increased from TA to TB and then to TC under SPD treatment, but had no significant changes from A to B then to C with water treatment. Among them, four tannin compounds (1,4,6-tri-O-galloyl-β-D-glucose, 1,3,6-tri-O-galloyl-β-D-glucose, 2,4,6-tri-O-galloyl-D-glucose, and 1,2,3,6-tetra-O-galloyl-β-D-glucose) were continuously increased from TA to TB and then to TC, and their relative abundances were higher than other phenolic acids at all stages after SPD treatment. Besides, the seven amino acids significantly increased with FI following SPD treatment and had no significant changes at all stages with water treatment); L-homoserine (the precursor of ethylene), L-serine, and L-threonine were the most abundant under the SPD treatment. In addition, the proline and its premise substances involved in the biosynthesis of amino acids presented a continuous increase

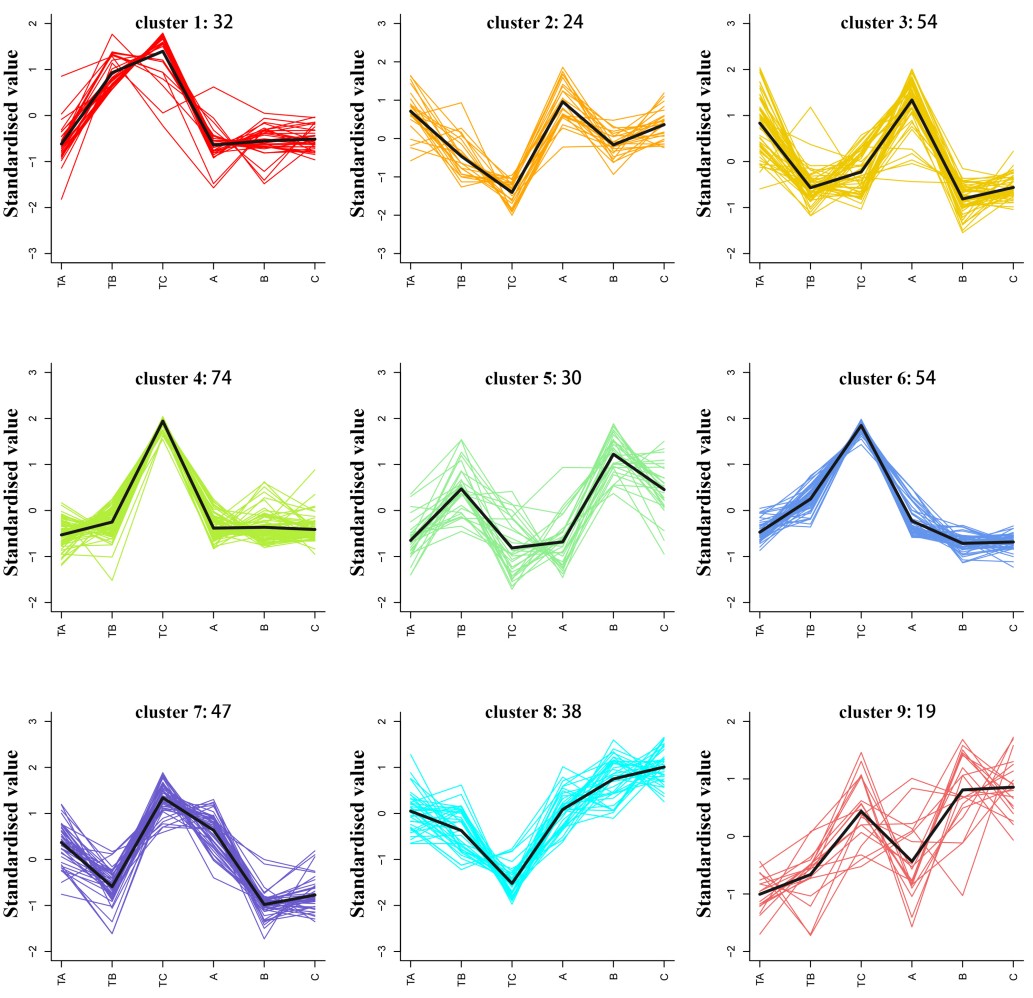

**Figure 6  K-means analysis of all differential metabolites.** Abscissa indicates the name of the sample, ordinate indicates the standardized relative amounts of metabolites, and the numeric characters following each cluster indicates the number of metabolites in this cluster.

(TA to TB to TC) following SPD treatment, however, showed no significant changes (A to B to C) after water treatment (Fig. 7B).

## Identification of carbohydrates and vitamins during FI after SPD treatment

Additionally, seven vitamins and 11 saccharides and alcohols showing significant changes between the TA and TC stages by SPD treatment were identified, and they showed different accumulation patterns in samples of different days after SPD and water treatment (Fig. 7C and Table S6). Among the 11 saccharides and alcohols, 10 carbohydrates, including D-fructose6-phosphate, D-fructose-1,6-biphosphate, glucose-1-phosphate, sorbitol-6-phosphate, and D-glucose 6-phosphate, showed a sharp increase from TA to TC by SPD treatment but no significant changes from A to C after water treatment. Among the 7 vitamins, erythorbic acid and L-ascorbic acid, with the highest relative abundance, showed

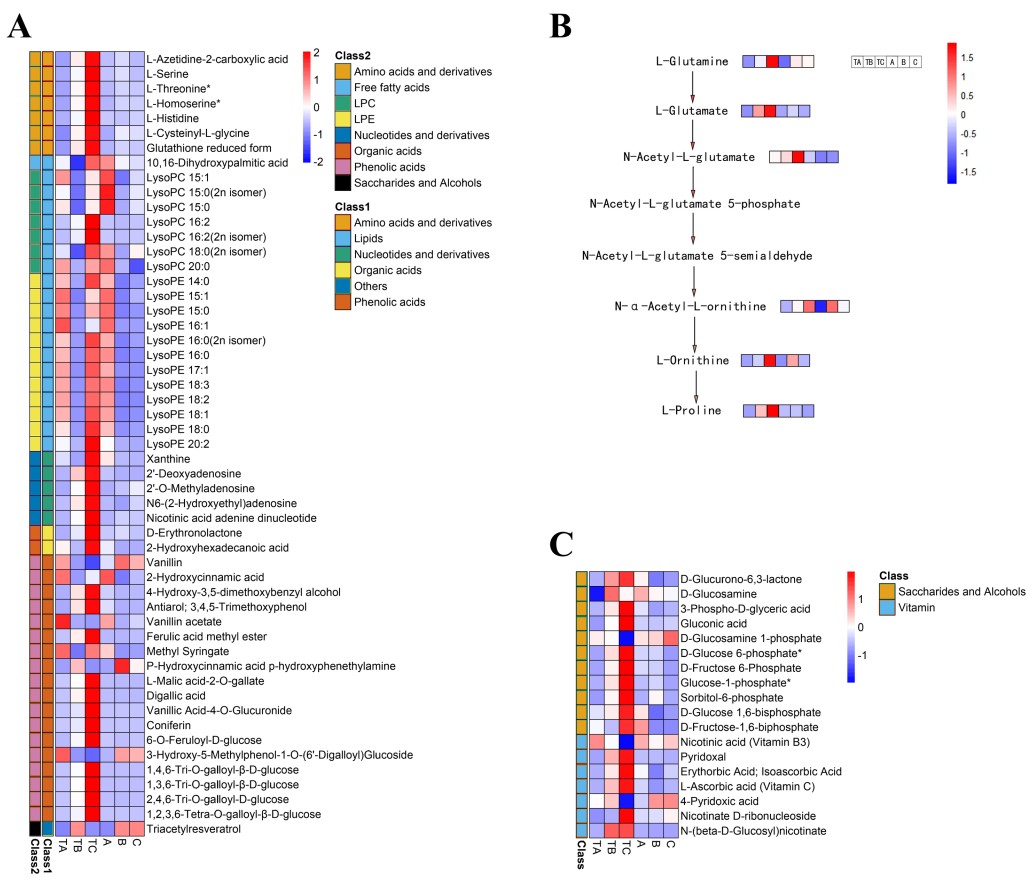

**Figure 7   Heat map of differential metabolite contents in the mango stem apex at different stages.** (A) Heat map of amounts of 53 differential metabolite content at three stages of FI (TA, TB, and TC); (B) levels of metabolites in proline biosynthesis pathway in mango stem apex at three stages of Fl. (C) Heat map of amounts of 18 differential metabolite (seven vitamins and 11 saccharides and alcohols) content at three stages of FI (TA, TB, TC, A, B and C); Note: heat map shows the level of metabolites in mango terminal buds at three stages, and blue to red in heat map shows the level of metabolites from low to high.

a substantial increase from TA to TC following SPD treatment, while no significant changes from A to C by water treatment. In addition, 14 out of 372 differential metabolites changed more than ten times at the different stages of FI treated with SPD, but had no significant changes treated with water (Fig. 8 and Table S7); this group included L-Homocysteine (TA *vs* TB, TA *vs* TC), L-Histidine (TA *vs* TC), and L-Homomethionine (TA *vs* TC, TB *vs* TC).

## DISCUSSION

Recently, the metabolic regulation of flower bud dormancy in temperate fruits has attracted the attention of numerous researchers (*Zhuang et al., 2015*; *Gabay et al., 2019*; *Yang et al., 2021*). The FI in mango, which is accompanied by the release of a dormant bud and regulated by low temperatures (*Núñez Elisea & Davenport, 1995*; *Wilkie, Sedgley & Olesen, 2008*; *Ramírez & Davenport, 2010*), seems to be similar to the regulation of flower bud

| | | | | | | |
|---|---|---|---|---|---|---|
| 0.30 | 0.56 | 0.26 | 9.98 | 10.68 | 0.70 | L-Homocysteine |
| -0.15 | 0.05 | 0.19 | 1.61 | 3.37 | 1.76 | L-Histidine |
| -0.34 | -0.16 | 0.17 | 0.47 | 4.57 | 4.09 | L-Homomethionine |
| -0.41 | -0.19 | 0.22 | 1.70 | 3.48 | 1.78 | Digallic acid |
| -0.25 | -0.62 | -0.36 | 3.52 | 6.74 | 3.22 | Vanillic Acid-4-O-Glucuronide |
| -0.34 | 0.09 | 0.42 | 4.86 | 3.98 | -0.88 | 1-O-Galloyl-2-O-Cinnamoyl-β-D-glucose |
| -1.13 | 0.05 | 1.19 | 4.64 | 5.39 | 0.75 | 2,3-Di-O-Galloyl-D-Glucose |
| -0.77 | -0.05 | 0.72 | 1.59 | 3.74 | 2.14 | LysoPC 16:2 |
| -0.75 | -0.12 | 0.63 | 1.58 | 3.93 | 2.35 | LysoPC 16:2(2n isomer) |
| -0.12 | -0.21 | -0.10 | 2.29 | 4.53 | 2.24 | 1,4,6-Tri-O-galloyl-β-D-glucose |
| 0.03 | -0.02 | -0.05 | 2.84 | 5.02 | 2.17 | 1,3,6-Tri-O-galloyl-β-D-glucose |
| -0.47 | 0.12 | 0.60 | 2.51 | 4.59 | 2.08 | 2,4,6-Tri-O-galloyl-D-glucose |
| 0.90 | 0.76 | -0.15 | 1.80 | 3.57 | 1.76 | Nicotinic acid adenine dinucleotide |
| -0.05 | -0.07 | -0.01 | 1.82 | 4.18 | 2.36 | 1,2,3,6-Tetra-O-galloyl-β-D-glucose |
| A vs B | A vs C | B vs C | TA vs TB | TA vs TC | TB vs TC | |

**Figure 8 Metabolites with relative abundance changed more than 10 times.** The abscissa is group, the ordinate is differential metabolites, and the numbers in the diagram relate to logarithmic value of differential multiples of differential metabolites based on two.

dormancy in temperate fruit trees. However, knowledge of the changes in metabolites in mango buds from dormancy to flower initiation is limited. The present study used UPLC-MS/MS-based widely targeted metabolomics to reveal the overall dynamics of the primary metabolites in mango bud at three representative stages of FI. A total of 582 metabolites were identified, of which 372 were differentially accumulated during the three FI stages. Metabolites including lipids, phenolic acids, amino acids, carbohydrates, and vitamins were further characterized. Notably, the analysis revealed that metabolites such as proline changed significantly during FI in response to SPD treatment. Thus, this study provides evidence for novel mechanisms underlying FI in mango.

In Zhanjiang, China, it is normal for some mango varieties to not blossom without any treatment, such as Tainong-1 and Renong-1. Based on our results on fruit yield, we found that the mango yield in 2021 was slightly lower than that in 2020. It may be caused by damage of insect pests, plant pathogens, soil and water pollution, poor pollination, stresses of water and temperature. Approximately 5–80% of mango yield losses in India are directly or indirectly caused by fruit flies (*Stonehouse, 2001*). In recent years, mango leaf, branch, and fruit infections by bacterial black spot have become increasingly serious in the mango-producing areas, resulting in lower yields and product quality (*Zhou et al., 2019*). There is global evidence that salinity pollution of soil and water reduces crop production (*Haque SAnwarul, 2006*). Pollination is vital for most fruit trees, and poor pollination leads to poor fruit quality and low yield (*Kleiman, Koptur & Jayachandran, 2021*). When crops are under-irrigated, they risk water stress, which can lead to a reduction in yield or quality (*Jabro et al., 2020*). Temperature is essential in inducing flowering in mango trees because

bud growth occurs at temperatures of around 10 °C at night and 20 °C during the day, resulting in flower buds (*Santos-Villalobos et al., 2013*).

The analysis of differential metabolites between different samples showed that the number of differential metabolites in buds gradually increased with time after SPD treatment, while the number in control buds gradually decreased. These results indicated that SPD treatment had an important effect on metabolites in mango buds, and the effect was most significant at 80 and 100 days after the treatment. The metabolism changes were consistent with the morphological development.

KEGG enrichment analysis revealed that floral induction of mango involved multiple metabolic pathways. *Zhang et al. (2022)* discovered that a higher amino acid content was beneficial to walnut flower induction. Rachapudi et al. discovered that biosynthesis of amino acids and 2-oxocarboxylic acid metabolism, carbon metabolism and fatty acid metabolism were also involved in floral induction of *Pongamia pinnata* (*Sreeharsha et al., 2016*). *Yan et al. (2022)* discovered that aminoacyl-tRNA biosynthesis was involved in floral induction of early-spring plants. This study indicated that multiple metabolic pathways may jointly regulate flowering induction in mango. Amino acid and lipids may play an important role. K-means clustering analysis of differential metabolites revealed that levels of phenolic compounds changed significantly during the 80–100 days following SPD treatment, but not in the 80–100 days following water treatment. It was demonstrated that phenolic acids promoted mango blossoming (*Srilatha & Reddy, 2015*; *Srilatha et al., 2016*). In this study, lipids levels increased significantly in the 80–100 days following SPD treatment, but not in the 80–100 days following water treatment. Research has proposed that endogenous lipids of mango shoots have hormonal activity and play an important role in mango flowering (*Kumar, Ram & Pant, 1989*). In this study, amino acids levels increased significantly in the 30–100 days following SPD treatment, but not in the 80–100 days following water treatment (*Shivashankara, Geetha & Roy, 2019*; *Nj et al., 2017*). The results showed these pathways respond to floral induction of mango (*Prates et al., 2021*).

Lipids are components of the cell membrane and play essential roles in regulating different biological pathways (*Quartacci et al., 1995*; *Hamrouni, Salah & Marzouk, 2001*). Lipids in plants can be divided into several categories, including glycerophospholipid (GP), glycerolipid (GL), and fatty acyl (FA). GPs are further divided based on the head group structures into the following classes: phosphatidylcholine (PC), phosphatidylethanolamine (PE), phosphatidylserine (PS), phosphatidic acid (PA), phosphatidylinositol (PI), phosphatidylglycerol (PG), and cardiolipin (CL). During bud dormancy in European pears, phospholipids significantly increased with chilling accumulation (*Gabay et al., 2019*). Significant escalations in the LPC and LPE content in the Arabidopsis leaves were observed in response to bacterial infection and low temperatures. Meanwhile, an increase in phosphocholine, the precursor for the biosynthesis of PC, was detected during flower bud dormancy release in black currant (*Jung et al., 2017*). In this study, 18 lipids changed significantly following SPD treatment, and LPE 17:1 and LPE 15:1 showed more than ten-fold change from 80 to 100 days after SPD treatment, highlighting their role in mango FI in response to SPD treatment. Interestingly, biochemical analysis in Arabidopsis showed that FLOWERING LOCUS T protein (FT) binds to diurnally changing molecular species
of PC to promote flowering, indicating a role of PC in the timing of flowering (*Nakamura et al., 2014*; *Nakamura et al., 2019*). However, the importance of lipid metabolism during FI treated with SPD treatment of mango needs to be further studied by integrating relative gene transcription analysis.

Phenolic compounds are another set of metabolites that play essential roles in regulating the growth and development of plants (*Halbwirth et al., 2006*; *Edwards et al., 2008*). Phenolic acids are organic acids with phenolic groups. The phenolic hydroxyl groups form stable phenoxy radicals with free radicals and eliminate free radicals in vivo. Wild chokeberries and cultivar "Viking" with higher phenolic acid content have intense anti-radical activity (*Jakobek et al., 2012*). *Oliveira et al. (2020)* proved the apparent accumulation of phenolic compounds in mango stem apex before flower bud initiation from the cytological point of view. In the present study, phenolic acids accounted for a high proportion of the metabolites that showed significant changes from 30 to 80 then to 100 days by SPD treatment. Most of them were continuously increased, which may be involved in free radical scavenging. Meanwhile, studies have proven that tannins and related compounds, including coumarin, trans-cinnamic acid, and many phenolic compounds, are gibberellin antagonists (*Corcoran, Geissman & Phinney, 1972*). Therefore, the up-regulated tannin compounds also probably antagonize gibberellin during flower bud differentiation to promote flowering.

The release of bud dormancy is accompanied by proline accumulation (*Seif El-Yazal, El-Yazal & Rady, 2014*). Besides, free proline accumulation is a typical response of plants to various stresses, maintaining the osmotic balance between cytoplasm and vacuole. Proline can remove reactive oxygen species produced under stress and be used as an osmotic agent to protect the subcellular structure (*Zhuang et al., 2015*). Studies have shown that the low levels of proline in switchgrass delay flowering. In addition, proline is involved in flowering signals (*Schwacke et al., 1999*; *Mattioli et al., 2009*; *Kavi Kishor & Sreenivasulu, 2014*; *Kim et al., 2021*). Transgenic arabidopsis overexpressing Δ1-Pyrroline-5-carboxylate synthetase (AtP5CS1) bloomed early and accumulated a large amount of proline under long-day and short-day conditions (*Mattioli et al., 2012*). AtP5CS2 is a multi-allelic AtP5CS gene, an early target of CONSTANS (CO), and participates in flower formation (*Samach et al., 2000*). Many differential metabolites were related to amino acid biosynthesis in this study, and L-proline and its precursors increased continuously during FI in response to SPD treatment (Fig. 7B), with high relative abundance per stage. These observations indicated that the dormancy release of mango buds followed proline accumulation.

Plants require carbohydrates for growth and development; they provide energy to maintain the growth of buds after dormancy and link sugar signals with flowering-related pathways (*Zhuang et al., 2013*; *Chen et al., 2018*). *Guillamón et al. (2020)* found that D-fructose-2-phosphate and D-fructose-2,6-biphosphate in almond flower buds increased at the stages close to endodormancy release. In this study, saccharides accumulated from 30 to 100 days of SPD treatment were mainly those substances related to starch and sucrose metabolism. These findings indicate that the accumulation of saccharides (D-fructose 6-phosphate, D-fructose-1,6-biphosphate, glucose-1-phosphate, and D-glucose 6-phosphate) may be important in mango flower bud development after SPD treatment (*Upreti et al.,*

*2014*). Meanwhile, D-glucosamine 1-phosphate alone was decreased. Previous studies have shown that glucosamine 6-phosphate may be metabolized to fructose 6-phosphate, increasing glycolysis flux (*Xing et al., 2008*). In red-rice seeds, endodormancy release was associated with increased glycolysis and energy obtained (*Gianinetti et al., 2018*). These findings are consistent with our results in mango stem apex, indicating the role of the carbohydrate in FI.

Furthermore, erythorbic acid and L-ascorbic acid were abundant and continuously accumulated vitamins from 30 to 100 days following SPD treatment, but no significant changes were observed in the water treatment. Ascorbic acid is an important antioxidant and an auxiliary factor for synthesizing few hormones (*Barth, 2006*; *Bilska et al., 2019*). It also links flowering time with response to pathogens through a complex signal transduction network (*Du et al., 2016*). In Japanese pear and sweet cherry, reactive oxygen species increased during endodormancy and decreased after endodormancy release (*Gianinetti et al., 2018*; *Baldermann et al., 2018*). Additionally, research has demonstrated that a rise in ascorbic acid levels with an increase in flavonols is implicated in the degradation of hydrogen peroxide ($H_2O_2$) in grapevine leaves (*Vitis vinifera* L.) (*Pérez, 2002*). Therefore, we speculate that ascorbic acid as an antioxidant can scavenge free radicals, and its accumulation promotes mango flower bud differentiation.

In addition to the substances mentioned above, a few other metabolites (Fig. 8), including L-Homocysteine, L-Histidine, and L-Homomethionine et al., may also play important roles in some specific stages. The relative abundance of these metabolites changed more than ten folds at a particular stage after SPD treatment. The significant changes in the relative abundance of these metabolites during FI in response to SPD treatment indicate their role in mango FI. Further analysis of the role of these metabolites may clarify the molecular mechanism of mango FI by SPD treatment.

## CONCLUSIONS

In this study, we confirmed the positive role of SPD in promoting mango flowering. Significant changes in the abundances of metabolites such as lipids, tannins, proline, ascorbic acid, and saccharides were detected in the buds from 30 to 80 and then to 100 days after SPD treatment, indicating their crucial role during mango FI in response to SPD treatment. The results will provide a novel approach to mango flowering management in subtropical regions and provide a theoretical basis for applying SPD to regulate mango flowering.

## ACKNOWLEDGEMENTS

The authors thank Wuhan Metware Biotechnology Co., Ltd., for the widely targeted metabolomics analysis. We acknowledge TopEdit LLC for the linguistic editing and proofreading during the preparation of this manuscript.

### Funding

This research was funded by the National Key R & D Program of China (No. 2020YFD1000604, 2019YFD1000500), the Hainan Province Natural Science Foundation of China (320QN323, 321MS076), the Guangdong Provincial Special Fund for Modern Agriculture Industry Technology Innovation Teams (2019KJ108), and the Project of Enhancing School With Innovation of Guangdong Ocean University (GDOU2013050217, GDOU2016050256). The funders had no role in study design, data collection and analysis, decision to publish, or preparation of the manuscript.

### Grant Disclosures

The following grant information was disclosed by the authors:
National Key R & D Program of China: 2020YFD1000604, 2019YFD1000500.
Hainan Province Natural Science Foundation of China: 320QN323, 321MS076.
Guangdong Provincial Special Fund for Modern Agriculture Industry Technology Innovation Teams: 2019KJ108.
Project of Enhancing School With Innovation of Guangdong Ocean University: GDOU2013050217, GDOU2016050256.

### Competing Interests

The authors declare that they have no conflict of interest.

### Author Contributions

- Fei Liang conceived and designed the experiments, performed the experiments, analyzed the data, prepared figures and/or tables, and approved the final draft.
- Wentian Xu conceived and designed the experiments, performed the experiments, analyzed the data, prepared figures and/or tables, and approved the final draft.
- Hongxia Wu analyzed the data, prepared figures and/or tables, and approved the final draft.
- Bin Zheng analyzed the data, prepared figures and/or tables, and approved the final draft.
- Qingzhi Liang analyzed the data, prepared figures and/or tables, and approved the final draft.
- Yingzhi Li conceived and designed the experiments, analyzed the data, prepared figures and/or tables, and approved the final draft.
- Songbiao Wang conceived and designed the experiments, analyzed the data, prepared figures and/or tables, and approved the final draft.

### Data Availability

The raw data are available as Supplemental Files.

## Supplemental Information

Supplemental information for this article can be found online at http://dx.doi.org/10.7717/peerj.14458#supplemental-information.

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
