# Peer review of "Widely targeted metabolite profiling of mango stem apex during floral induction by compond of mepiquat chloride, prohexadione-calcium and uniconazole"

_PeerJ, doi:10.7717/peerj.14458_

## Round 0.1 · original submission · Major Revisions

I strongly agree with all three reviewers. This article requires suggested changes by reviewers before accepting for publication.

Reviewer 1 ·

Basic reporting

Liang et al., investigates the effect of SPD (a floral induction agent) on the metabolomic landscape in terminal buds of mango tree. The authors uncovered a wide range of metabolic changes mainly on secondary metabolism upon SPD treatment. The authors concluded that the metabolic data present in this study provides a knowledge basis for understanding how SPD induces flowering. Overall, I find the study to be well conceptualized. However, I do have several major questions and suggestions on the manuscript, as well as some minor points.

Experimental design

Main point 1. SPD is proposed as a safer treatment to induce mango FI, compared to PBZ. I would suggest the same metabolomics experiments on terminal buds from PBZ-treated group.

Main point 2. What is the rationale for choosing day 30, 80, and 100 post-SPD/water treatment?

Validity of the findings

Main point 1. Results on metabolomics need to be more organized. The following questions need to be addressed: 1) How many metabolites are constantly altered between SPD treated vs mock treated across three time points? 2) How many metabolites change between time points?

Main point 2. PCA. What is QC? I see a nice time dependent trend that SPD treatment has an increasing effect on metabolome as time prolongs and maximizes at the time point C (100 days post treatment).

Main point 3. Is it known how SPD is metabolized in plant cells? If yes it would be beneficial to review such pathways, because it would put the SPD-induced metabolomic changes in a more comprehensive context.

Additional comments

Minor point 1. I suggest the authors use full name of SPD in the manuscript title and remove content in the parenthesis. Titles are typically free of parenthesis.
Minor point 2. Line 57. Does inappropriate doses of PBZ mean too high or too low? It would be nice if the authors can point out if any studies have linked PBZ to human health.
Minor point 3. Line 109. Typo in eighteen. Line 39 Typo in utilizing.
Minor point 4. Line 216-217. “And give the sample a name.” This is not a complete sentence.
Minor point 5. What does CK stand for in Figure 1? Statistical significance should be added.

Reviewer 2 ·

Basic reporting

1. There are many small mistakes need carefully re-check before publication, such as explanation of abbreviations, wrong sample/treatment name, spelling, etc.
2. For Figure 3, 4, 8, and 10, you should present the mean of repetitions/various blocks under each treatment like what you have done in other figures.
3. More necessary details are need in Materials and Methods, please check the comments in next part.
4. I didn't see the objective of doing data analysis in Figure 4, 5, 6, 7. These figures were not explained well in results and discussion. Please evaluate if they are necessary in this article.

Experimental design

Line 27: Use ‘,’ instead of ‘,’. You should mention Variable Importance in Projection is shorted as VIP, the same for FC and other abbreviations. Or else, you may make a table explaining all the abbreviation used in this article.
Line 29: treatments
Line 33, 109: check the spelling
Line 43: Italicize scientific names, also check other scientific names
Line 44: regions (He et al., 2021). China is the second largest mango production country …
Line 50: low temperature instead of cold temperature
You didn’t mention gibberellin was shorted as GA, please carefully check all abbreviations, such as in line 100-101, you should use UPLC-MS/MS.
Line 67: ‘activity’- do you mean effects?
Line 66-71: Please add reference source.
Line 80: in vegetative buds, check other English writings in this line.
How about the climate (temperature, precipitation, etc.) in the orchard? Did you prune the trees before the FL? How about the planting density, irrigation, and fertilization condition in the orchard? Where did you spray the SPD, on foliage or to the soil? How did you guarantee the nine trees were sprayed equally with 12 L SPD? Did you apply SPD/water for only once? How were the pollination requirements being satisfied?
Line 121-122: Stage 1 is the dormant stage when buds were dormant, referring to the 60 days after SPD/water treatment.
Line 123: Do you mean the stage 2 is referring to 60-80 days after SPD/water treatment?
Line 124: Although the ‘writing brush head’ gave me a lively imagination, I didn’t find this phrase in bud description. You may find a more professional alternative here.
Line 132: How did you count the branch number on a tree? Did you count the number of panicles for one time or many times?
Line 138: The producer location of lab equipment needs to be mentioned, also check for all other equipments.
Line 149-151: The mobile gradient program description is not accurate. Either use 0, 9, 10, 11.1, and 14 min in present description, or explain as ‘%A, %B for 9 min, then %, % for 1 min…’
Line 177: Since you have mentioned the version of R software before, you do not have to repeat the version number in the following.
Line 189: We analyzed the flowering rate… Please explain ‘flowering rate’.
Line 192-193: The number of harvested fruits can not be found in the data you presented (Fig. 1 or Table S1).
Line 194: What objective factors do you imply here? This should be included in discussion part instead in the results.
Figure 2: T-30/C-30 indicates mango stem apex at 30 days after SPD/water treatment.
Line 208: loosened
Line 216: Please rewrite the last sentence of this paragraph.
Table S2: What is mix01, mix02, and mix03?
Figure 3A: What is QC (purple dots)?
Line 220-232: For PCA result, can you briefly describe the variables with main contributions to PC1 and PC2? For HCA result, can you briefly describe the metabolic characteristics in the six distinct groups?
Line 235: A and TA or A and TB?
Line 239-248: In Figure 5, please explain up regulated / down regulated. I guess up regulated means higher concentration of the metabolites in SPD treated buds than in control. However, you use C vs TC, which implies control / SPD, then up regulated means higher concentration in control than in SPD treated buds.
Line 258-260: How did you get these conclusions?
Line 260-268: Again, since you use A vs TA, B vs TB, and C vs TC in figure 6, the definition of the rich factor is not clear. It seems all pathways included in Figure 6B&6C are significantly different (P<0.05) between B and TB, C and TC. How did you get the conclusion from line 263 to 268? Besides, it is necessary to explain more information in Fig. 6 here.
Line 271: What is the meaning of ‘average value’ here, average of which variables?
Line 276-280: You may use ordinal numbers instead of cardinal numbers in current description. Or you can express it as ‘cluster 2, 3, 4, 6, and 8.’
Line 271-281: What is the purpose of the K-means clustering analysis? If you summarized some tendency among the six treatments in different clusters, you need to explain the tendency in your results.
Figure 7: Please capitalized the first letter in a sentence. Please carefully check small mistakes in English writing of the whole article.
Line 284-291: The selected 53 metabolites might be related to FI instead of SPD treatment.
Line 286: FI or F1?
Line 291-292: Explain the meaning of the numbers within brackets.
Line 328: Delete ‘, a tropical fruit crop,’.
Line 335: FI or F1? Please carefully check the whole article.
Line 341-343: Please carefully check 'GP' and rewrite this part.

Validity of the findings

Only 18 mango trees were included in this study, the sample size is not big enough to exclude outliers. But for the flower formation rate and yield data, the experiment was repeated for two years, which is a good supplement for a small sample sized experiment design. For the control groups (water) in metabolites analysis, flower induction did not occur on 8 out of 9 trees in 2021 (Table S1). You need to discuss the reasons.

Reviewer 3 ·

Basic reporting

In the present manuscript written by Liang et.al, authors have tried to study the effect of SPD on mango floral induction by metabolite profiling. The problem which has been addressed by the authors is
unique and authors have tried to filled the knowledge gap by metabolite profiling. Although, major revision is required in the current form of manuscript. Following points could be considered

The whole language of the article needs a significant modification. If possible then authors need to add more recent references such as in line 81.
Figures3 and 4 can be combined into a single figure. Figures 5 and 6 can be combined together. Figures 8,9 and 10 can also be combined in a single figure and Figure 11 could be added to supplemental information.
In table S2 any columns have text from other language script, as the targets readers are international readers, authors could change the language of those columns to English for better readability.
All the figure legends are needed to be more detailed.

Experimental design

Although authors have described the sample preparation but it’s not clear in the manuscript about the analysis of blank samples to verify the absence of interfering species at the retention time of the analytes. Also, it is not clear that were the bud samples weighed/ measured? How were the detected metabolites normalized?
Authors should also added about the standard solutions used in the study. If authors performed any optimization of UPLC conditions, then adding that information will help readers.
Under the method section, while describing about UPLC MS/MS system, information regarding ion transitions should be added, how many ion transitions were used for each compound. The methods are required to be written in more detail.
In order to test the validity of optimized conditions have authors repeated the assay, if yes, the authors are required to add that information.
Additionally, the selectivity, linearity, limits of quantitation (LOQs), limits of detection (LODs), precision, accuracy, and stability of the metabolite detection are needed to be added.

Validity of the findings

Under results section in Fig 1, authors have shown the flowering rate and yield per tree but authors are required to perform the statistical analysis with pairwise comparisons and indicate the p values to show the significance. Individual values along with SEM and error could also be plotted that way authors would not have to include Supplemental table S1.
All the statistical analysis are also required to be briefly added in the figure legends.
In Figure 4, information about metabolites are needed to be added.
Link to the KEGG database should be provided.
Overall the conclusions are required to be rewritten which indicated the key findings of metabolite profiling. The overall impact of SPD on key metabolites expression and how the results from the present study could be implicated in the field.

Additional comments

Line 58-59 needs to be rewritten. Incomplete sentence, line 82 “decreased delayed flowering”
Line 91 development and reproduction.
Line 117-118, incomplete, broken sentences.

---

## Round 0.2 · Minor Revisions

The manuscript still needs to be consulted with a colleague who is an expert in the English language or needs to get professional help in drafting the final version. The manuscript can be accepted after the English language correction.

Reviewer 1 ·

Basic reporting

Liang et al., investigates the effect of SPD (a floral induction agent) on the metabolomic landscape in terminal buds of mango tree. Most of my comments have been adequately addressed by the authors.

Experimental design

The study is well conceived and aligned with the aims and scope of the journal. Statistical analyses used in the study are rigorous and of good quality. The methods section is described with sufficient details after the authors' revision (addition of UPLC-MS-MS method details, clarification of QC samples, and revised figure captions).

Validity of the findings

In my opinion this manuscript has been significantly improved but there are still a few omissions that would benefit from being addressed.

Line 220. If I am understanding correctly, SPD treated group show a slight decrease in mango yield per tree comparing to the control. Does this observation make SPD unfavorable as a potential flower induction reagent?

Colloquial expressions such as don’t (Line 127) and can’t (Line 63) should be replaced with do not and cannot.

Line 365. The ‘et al’ at the end of the sentence does not make sense.

Additional comments

No comment

Reviewer 2 ·

Basic reporting

Most of my comments were responded properly by authors. I am OK with these revisions. However, too many typos can be found in the whole manuscript, especially in the revised part. The English language needs extensive improvement before publication.

Experimental design

no comment

Validity of the findings

no comment

Additional comments

For figures and tables, please add necessary legends.

There are still a lot of typos. English needs to be improved. For example, line 18, 29, 30, 63, 64, 66, 139, 141, 153-154, 168-173, 247, 300, and so on.
Line 142: SPD is a ‘homemade’ exogenous GA inhibitor? Is the formula created by growers or researchers?
Line 143: ‘Except for a brief temperature drop in mid-December (below 15℃ for about one week), the average monthly temperature of other months is higher than 15℃. Before flowering, branches were not pruned, and SPD was sprayed on leaves. In our pre-experiment, we sprayed about 12L water on the leaves of 9 trees. We only used SPD once, and flies were bred in the garden to pollinate.’ I didn’t find this information was added to materials and methods. Temperature, precipitation, climate, and cultivation management (irrigation, fertilization, planting density, etc.) are essential in materials and methods. It is important for audiences to repeat, continue, or cite this study.
Line 181-182: The website of the company is not necessary. Besides, starting from the 2nd time a company emerges in the manuscript, the location does not need to be mentioned again.
Lines 260 & 263: The ‘writing brush head’ was not revised in these 2 lines.

Reviewer 3 ·

Basic reporting

Authors have incorporated the suggested changes.

Experimental design

No comments

Validity of the findings

No comments

Additional comments

No comments

---

## Round 0.3 · accepted · Accept

We appreciate your effort in further improving your manuscript; Your paper is a valuable contribution to plant metabolomics.